# Spectroscopy-Based Evaluation of the Antioxidant Capacity of Hemp (*Cannabis sativa*)

**DOI:** 10.3390/ijms262311696

**Published:** 2025-12-03

**Authors:** Mariola Bartoszek, Justyna Polak, Paweł Gała, Michał Zieliński, Krzysztof Nawrot, Mirosław Chorążewski

**Affiliations:** 1Institute of Chemistry, University of Silesia, Szkolna 9, 40-006 Katowice, Poland; miroslaw.chorazewski@us.edu.pl; 2Institute of Law, Faculty of Law and Administration, University of Silesia, Bankowa 11 b, 40-007 Katowice, Poland; 3The Professor Tadeusz Widła Interdisciplinary Research Centre for Forensic Science and Legislation, The University of Silesia in Katowice, Bankowa 11b, 40-007 Katowice, Poland; 4Toplanta Sp. z o.o., Kościelna 6, 63-645 Siemianice, Poland

**Keywords:** functional foods, hemp, antioxidant capacity, EPR spectroscopy, TEAC, TPC

## Abstract

In the present study, a comprehensive evaluation of the antioxidant properties of various parts of the hemp plant, specifically its leaves and flowers, in a new feminized variety of *Cannabis sativa* with an admixture of *Cannabis ruderalis* was investigated. Methods such as Electron Paramagnetic Resonance (EPR) and UV-visible (UV-vis) spectroscopy were used to assess trolox equivalent antioxidant capacity (TEAC) and total polyphenol content (TPC). TEAC values of the analyzed samples ranged from 29.5 μmol TE/g DW to 150.2 μmol TE/g DW, while TPC varied between 5.4 mg GAE/g DW and 20.3 mg GAE/g DW. The findings indicate that hemp exhibits significant antioxidant properties, especially in its leaves. This is attributed to a diverse range of antioxidants, including cannabinoids, flavonoids, and phenolic compounds, which offer notable health benefits. Furthermore, the method of drying hemp has been shown to influence its antioxidant properties significantly. Research indicates that freeze-dried and air-dried hemp retains higher levels of antioxidant compounds compared to other drying methods. This suggests that selecting an appropriate drying technique is essential for preserving the bioactive compounds responsible for hemp’s antioxidant activity.

## 1. Introduction

In recent years, products with antioxidant properties have garnered increasing interest in dietetics, protective therapy, and the treatment of various diseases. Due to their bioactive ingredient content, such food products are classified as functional foods in the literature of food law [1,2]. Given the coexistence of many different definitions of the concept in both legal and natural sciences, the European Commission funded an initiative in the late 1990s to establish a scientific approach to studying functional foods. The EC Joint Action for Functional Food Science in Europe (FUFOSE) involved numerous European experts in nutrition and related sciences, producing a report that serves as a widely used foundation for further discussion and the evolution of the functional food concept [3]. This report proposed a working definition of functional foods as those that beneficially affect one or more target functions in the body beyond their nutritional effects in ways that are important for improving health and well-being and/or reducing the risk of disease. Since many antioxidant compounds are not synthesized endogenously, it is essential to include functional foods in the diet, particularly for combating free radicals.

Consequently, numerous studies have focused on foods and beverages rich in antioxidants such as vitamins, flavonoids, catechins, and other natural compounds, which can help prevent disease. Significant effort has been devoted to quantifying phenolic compounds in these products and exploring the relationship between their levels and the overall antioxidant properties of foods [4,5,6,7]. Antioxidants in foods and beverages play a vital role in the body’s defense against reactive oxygen and nitrogen species, commonly known as free radicals. These highly reactive molecules can damage essential biomolecules—including proteins, lipids, and DNA—thereby contributing to cardiovascular diseases, cancers, and age-related degenerative processes. By quenching the initiation and propagation of auto-oxidation chain reactions, antioxidants help mitigate oxidative stress. It is noteworthy that most natural antioxidants, such as vitamins, carotenoids, and flavonoids, are derived from plants, and their abundance in fruits and vegetables has been well documented [8,9,10]. Hemp *Cannabis sativa* L. (*Cannabaceae*) is renowned for its rich content of bioactive compounds, including cannabinoids, terpenes, flavonoids, and polyphenols, which contribute significantly to its antioxidant properties. These compounds have attracted considerable interest for their potential health benefits, such as anti-inflammatory, neuroprotective, and anticancer effects. Importantly, the drying method applied to hemp can markedly influence the retention of these bioactive compounds and, consequently, its antioxidant capacity.

Cannabis plants, in general, owe their antioxidant capabilities to compounds like polyphenols, terpenes, and cannabinoids [11]. Among these, delta-9-tetrahydrocannabinol (THC) and the nonpsychoactive cannabidiol (CBD) are the most recognized. Both THC and CBD exhibit antioxidant activities comparable to vitamins E and C [12], effectively scavenging free radicals, reducing metal ions, and counteracting oxidative stress [13]. Studies on rat neuronal cell cultures have demonstrated that their protective effects against oxidative damage can rival those of ascorbate and vitamin E [14]. Moreover, cannabinoids act as indirect antioxidants by modulating the redox balance, regulating glutathione (GSH) levels, activating antioxidant enzymes, and inhibiting prooxidant enzymes [14].

Various analytical methods have been developed to assess the antioxidant activity of fruits, vegetables, and herbs. Two primary mechanisms underpin these evaluations: hydrogen atom transfer (HAT) and single electron transfer (SET). For instance, the 2,2-diphenyl-1-picrylhydrazyl (DPPH) assay employs both HAT and SET mechanisms, though it predominantly relies on electron transfer reactions. In these assays, antioxidant activity is quantified by measuring changes in a stable free radical reagent following its interaction with antioxidants. Common screening techniques include oxygen radical absorbance capacity (ORAC), total radical-trapping antioxidant parameter (TRAP), photochemiluminescence (PCL), chemiluminescence (CL), total oxidant scavenging capacity (TOSC), total antioxidant capacity (TAC), and Cu(II)-reducing antioxidant capacity (CUPRAC) [15,16,17,18]. The results are typically expressed as Trolox Equivalent Antioxidant Capacity (TEAC).

Recent research has extended these investigations to hemp, focusing on the quantitative evaluation of its antioxidant properties. Studies have examined the total polyphenol and flavonoid content in hemp extracts, including measurements of CBD, THC, and IC50 values, which indicate the concentration of antioxidants required to reduce free radicals by half [19,20]. Additionally, antioxidant activities have been assessed using method such ferric reducing antioxidant power (FRAP) in various hemp varieties and under different cultivation conditions, as well as in extracts and oils with varying CBD/THC ratios. While these studies consistently confirm the antioxidant potential of hemp, none have yet employed electron paramagnetic resonance (EPR) spectroscopy—the only technique capable of directly detecting free radicals—to determine its total antioxidant capacity [20,21,22,23]. Unlike spectrophotometric assays, which rely on indirect measurements based on colorimetric or redox reactions, EPR provides direct and highly specific evidence of radical scavenging activity. This makes it a more accurate and reliable approach for evaluating antioxidant capacity, minimizing the risk of interference from non-radical reactions [24,25]. The principles of this approach are reported earlier in detail: in particular, the work [26] demonstrates and discusses the explicit plot of a good linear correlation between EPR-based and spectroscopy-based values of the antioxidant capacity, and Ref. [27] contains an illustrated explanation of the EPR-based method.

Given the importance of preserving antioxidant compounds during processing, this study investigates the impact of different drying methods on the antioxidant properties of hemp (*C. sativa*) leaves and flowers. Specifically, the research compares conventional air-drying, freeze-drying, and laboratory convection drying regarding their effects on total phenolic content, flavonoid content, and radical scavenging activity. By elucidating how drying techniques influence antioxidant retention, the study aims to optimize processing methods to maximize the nutritional and therapeutic potential of hemp-based products.

## 2. Results and Discussion

### 2.1. Study of Antioxidant Properties of Different Parts of Hemp

This study evaluated the antioxidant capacity (TEAC) and total phenolic content (TPC) of *C. sativa* extracts prepared from various plant parts: flowers, leaves, and their crushed counterparts dried using four distinct methods. The assessments were conducted using spectrophotometric UV-Vis and electron paramagnetic resonance (EPR) spectroscopy techniques (Table 1 and Table 2).

Our findings indicate that all *C. sativa* extracts exhibit notable antioxidant properties (Table 1 and Table 2).

The total antioxidant capacity of the analyzed samples ranged from 29.5 μmol TE/g DW to 150.2 μmol TE/g DW, from 22.5 μmol TE/g DW to 125.7 μmol TE/g DW, and from 36.7 μmol TE/g DW to 199.4 μmol TE/g DW (for TEAC_DPPH-EPR_, TEAC_DPPH-UV-vis_ and TEAC_ABTS-EPR_ respectively), while the total phenolic content (TPC) varied between 5.4 mg GAE/g DW and 20.3 mg GAE/g DW (Table 1 and Table 2). Overall, hemp exhibits antioxidant properties comparable to those of various herbs and spices [20,28,29,30]. The TEAC values determined for hemp (Table 1) exceed those recorded for several herbs, including mint (106 μmol TE/g DW), heartsease (32.5 μmol TE/g DW), common nettle (47 μmol TE/g DW), horsetail (86 μmol TE/g DW), chamomile (50 μmol TE/g DW), sage (128 μmol TE/g DW), fennel (10 μmol TE/g DW), and thyme (74–137 μmol TE/g DW). However, hemp demonstrates lower TEAC values compared to eyebright (506 μmol TE/g DW), St. John’s wort (396 μmol TE/g DW), and lemon balm (356 μmol TE/g DW) [28,29,30]. Regarding total polyphenol content, the values obtained for hemp are relatively high compared to some commonly studied herbs. Hemp contains more polyphenols than thyme (3.4–15.4 mg GAE/g DW), heartsease (18 mg GAE/g DW), mint (15 mg GAE/g DW), common nettle (16 mg GAE/g DW), horsetail (19 mg GAE/g DW), chamomile (8.5 mg GAE/g DW), and fennel (3.4 mg GAE/g DW). However, its TPC remains lower than that of highly polyphenol-rich spices and herbs such as cloves (150–250 mg GAE/g DW), oregano (30–100 mg GAE/g DW), rosemary (20–80 mg GAE/g DW), cinnamon (30–60 mg GAE/g DW), lemon balm (63 mg GAE/g DW), sage (32 mg GAE/g DW), eyebright (58 mg GAE/g DW), and St. John’s wort (45 mg GAE/g DW) [20,28,29,30,31]. These findings highlight hemp’s potential as a valuable source of antioxidants and polyphenols, placing it among the more bioactive plant-based materials commonly used in herbal medicine and functional food applications. The antioxidant potential of hemp is primarily attributed to its rich content of bioactive compounds, including polyphenols, cannabinoids, flavonoids, and other phytochemicals. These compounds act through various mechanisms, such as free radical scavenging, metal chelation, and inhibition of oxidative enzymes [32,33].

Furthermore, our study demonstrates that the preparation method of the plant material significantly influences the antioxidant properties. Specifically, extracts from crushed leaves and flowers exhibited higher TEAC and TPC values than those from uncrushed samples (Figure 1 and Figure 2). This suggests that crushing enhances the availability of bioactive compounds during extraction, thereby augmenting the antioxidant potential of the extracts. Moreover, the average values of TEAC determined by all applied methods, as well as the total phenolic content (TPC) (Table 1 and Table 2), were higher in the aqueous leaf extracts compared to those obtained for the flower extracts.

These findings underscore the importance of plant part selection and preparation methods in optimizing the extraction of antioxidants from *C. sativa*. They also highlight the potential of cannabis leaves, often considered a byproduct, as a valuable source of natural antioxidants.

The obtained Pearson correlation coefficients between total polyphenol content (TPC) and total antioxidant capacity (TEAC) are high ranging from 0.97 for TEAC_DPPH-Uv-vis_/TPC to 0.99 for TEAC_DPPH-EPR_/TPC and TEAC_ABTS-EPR_/TPC. It should also be emphasized that the correlation coefficients between the TEAC values determined using different methods and different radicals are high and amount 0.99 for TEAC_DPPH-EPR_/TEAC_ABTS-EPR_, TEAC_DPPH-EPR_/TEAC_DPPH-Uv-vis_ and TEAC_DPPH-Uv-vis_/TEAC_ABTS-EPR._ These high values indicate a strong positive correlation between all measured parameters and confirm the consistency and reliability of the results. EPR spectroscopy proves to be a highly suitable and reliable method for assessing antioxidant capacity, as it directly measures the presence of free radicals, providing an accurate reflection of radical-scavenging activity.

### 2.2. The Study of the Drying Method Effect on the Antioxidant Properties of Hemp

Hemp is a versatile plant known for its nutritional value and therapeutic properties. The drying process is essential in preserving the active compounds in hemp, particularly antioxidants.

The drying method employed in processing hemp significantly influences its antioxidant properties and overall nutritional value [34,35,36]. Our research indicates that freeze-drying and air-drying are superior techniques for preserving the active compounds in hemp compared to a laboratory convection dryer (Table 1 and Table 2, Figure 1 and Figure 2).

Freeze-drying is particularly effective because it removes moisture from the hemp at low temperatures, which helps to protect heat-sensitive compounds like polyphenols and cannabinoids from thermal degradation [36]. This method not only maintains the integrity of these compounds but also enhances their extraction efficiency. The formation of ice crystals during freeze-drying disrupts the cellular structure of the hemp, facilitating the release of phenolic compounds and making them more accessible for extraction with solvents.

On the other hand, laboratory convection dryer has been shown to cause a significant reduction in the antioxidant activity of hemp. The high temperatures (30 and 40) involved in the convection dryer can lead to the degradation of sensitive compounds, resulting in a loss of polyphenolic content and overall antioxidant capacity.

In conclusion, for maximizing the nutritional and therapeutic benefits of hemp, choosing freeze-drying or air-drying over a convection dryer is crucial. These methods not only help preserve the valuable antioxidant properties of the plant but also enhance the extraction potential of beneficial compounds, making them preferable options for processing hemp.

Air drying might be more financially viable for smaller operations with limited budgets, while freeze-drying may offer better quality products and higher profit margins for larger operations willing to invest in the necessary technology. The choice ultimately depends on the scale of production, available resources, and market demands.

## 3. Materials and Methods

### 3.1. Chemicals and Equipment

DPPH* (Sigma-Aldrich, Poznań, Poland) was used as the source of free radicals. To quantify the antioxidant capacity of the sample, trolox (Acros Organics, Geel, Belgium) was used. To determine total phenolic content, the Folin–Ciocalteu’s phenol reagent and gallic acid (POCH, Gliwice, Poland) were used. All other chemicals and solvents were of analytical grade and used without further purification.

Electron paramagnetic resonance (EPR) spectra were obtained with a Bruker EMX EPR spectrometer (Bruker-Biospin, Rheinstetten, Germany) operating at the X-band frequency (9.8 GHz) at room temperature. Typical instrument parameters were central field, 3480 G; modulation amplitude, 2.0 G; time constant, 40.96; gain, 1 × 104 G; microwave power, 20.12 mW; and sweep width, 60 G. UV–vis spectra were obtained with UV–vis spectrophotometer (Perkin Elmer, Shelton, CT, USA).

### 3.2. Samples

Fresh hemp from a plantation located in Siemianice, Poland (GPS: 51.179858, 18.137977) was used for the study. The plantation was established in a row system designed to ensure uniform sunlight distribution to all plants. Foils were used to suppress weeds in the large cultivation area, eliminating the need for herbicides. The varieties cultivated had a dominant *C. sativa* profile, with a slight admixture (5–15%) of *C. ruderalis* characteristics. The presence of sativa characteristics results in more sprawling, less compact inflorescences, which reduces the risk of flower rot and allows for better air circulation. These plants are characterized by continued growth into the flowering phase, although this growth slows over time. The addition of ruderalis characteristics ensures autophotoperiodicity—flowering is triggered by the plant’s age rather than the length of the day, as seen in photoperiodic varieties. To ensure sexual homogeneity within the cultivated population, feminized seeds were used. The feminization process involved treating female plants with substances that inhibit ethylene production, which prompted the female plants to produce pollen. By pollinating other female plants, almost exclusively female offspring were obtained.

The plants were harvested at two different times—on the 30th and 45th day of flowering. The harvest occurred at an advanced stage of flowering, but before it was completed, thus avoiding the typical “blooming” seen in seed crops, which could reduce the concentration of active ingredients in the flower biomass. The harvest was carried out manually; entire plants were cut and transported for drying using various methods.

The samples were divided into three groups corresponding to different drying methods: air drying, freeze drying, and drying in a laboratory convection dryer. In the freeze dryer and convection dryers, flowers and leaves were separated from the stem and then dried.

The drying process was performed using three different methods:Freeze drying (Sample S1): Flowers and leaves separated from the stem were dried in a freeze dryer (Lyophilizer Alpha 1–4, Christchurch, Germany) at −40 °C and 0.133 mbar for 3 days.Air drying (Sample S2): Stems with leaves and flowers were hung in a location with limited light and good air circulation and then dried naturally at 22 °C for 7 days. Relative humidity in the laboratory was maintained at approximately 40–45%. After drying, flowers and leaves were separated from the stem for further analysis. Convective drying (Wartmann, Utrecht, The Netherlands) at 30 °C (sample S3) and 40 °C (sample S4): Flowers and leaves separated from the stem were spread on white sheets of paper in a convection dryer and dried at the specified temperature for 3 days. The batch density in the dryer ranged from 0.5 to 1 kg/m^3^, and the airflow velocity was from 1 to 2 m/s. All samples were dried to a constant weight. In each case, the drying time was carefully selected to ensure that the dried materials achieved a similar water content (approximately 10%). Moisture content was measured using a moisture analyzer (Radwag, Radom, Poland). Some of the dried flowers and leaves were crushed and sieved through a 1 mm mesh sieve. The analyses were performed for dried leaves, dried flowers, leaves crushed and sifted through a 1 mm sieve, and flowers crushed and sifted through a 1 mm sieve.

Aqueous extracts were prepared by pouring 50 mL of water at 90–95 °C over 2 g of dried hemp material for 15 min. Extracts were obtained both from whole flowers (sample 1) and leaves (sample 2), as well as from crushed samples (sample 3 and sample 4) (particle size ~1 mm) that were sieved before extraction (see Table 1).

The total polyphenol content (TPC) and the Trolox equivalent antioxidant capacity (TEAC) were determined for all samples (see Table 1). Hemp was harvested twice, and the results presented are the averages of three measurements for each harvest.

### 3.3. Determination of Total Phenolic Content TPC

The total phenolic content of samples was determined using the Folin–Ciocalteu’s reagent, according to the spectrophotometric method described previously [25]. Briefly, 200 µL of Folin–Ciocalteu’s phenol reagent was added to 40 µL of sample diluted with 3.16 mL of distilled water. After 5 min, 600 µL of sodium carbonate solution was added. Samples were mixed and incubated for 30 min at 40 °C. The absorbance was read at 765 nm using a UV–vis spectrophotometer (Perkin Elmer, USA). A standard curve was prepared using gallic acid solution, and the regression equation was assessed as y = 1.04x + 0.006, where y is the absorbance and x is the concentration of gallic acid [mg/L].

### 3.4. Determination of Antioxidant Capacity TEAC Using EPR Method (TEAC_DPPH-EPR_)

Trolox equivalent antioxidant capacity (TEAC_DPPH-EPR_) was determined using the method described previously [21,24,25]. The regression equation for the linear relationship between the percent inhibition of EPR signal intensity and the mol number of trolox was assessed as %I = 1547.80C + 4.7, where %I is the inhibition and C is the concentration of the sample (volumetric percentage of the sample added relative to the total volume). This equation was used to calculate the antioxidant capacity of the studied samples in µmol TE per 100 mL of the studied sample. The percent inhibition of the EPR spectrum was calculated according to the following equation: %Inhibition = [(I_0_ − I)/I_0_] × 100%, where I_0_ is the area of the EPR spectrum of DPPH (control sample 200 µM DPPH solution), and I is the area of the EPR spectrum of DPPH with the sample.

For all samples, the regression equation describing the linear relationship between the percent inhibition (%I) of the EPR signal intensity and the concentration of the sample C was determined. Using this equation, the %I corresponding to C_100_ (the concentration equivalent to 100 mL of the sample) was calculated (I_100_). Then, based on the standard curve obtained for trolox, the antioxidant capacity was determined in µmol TE per 100 mL of the sample, and then recalculated per gram of dry matter used to prepare the extract. The antioxidant capacity measured using the DPPH method by EPR spectroscopy is expressed as TEAC_DPPH-EPR_.

EPR spectra were obtained with a Bruker EMX EPR spectrometer (Bruker-Biospin, Karlsruhe, Germany) operating at the X-band frequency at room temperature. The typical instrument parameters were central field, 3480 G; modulation amplitude, 2.0 G; time constant, 40.96; gain, 1 × 104 G; and microwave power, 20.12 mW.

### 3.5. Determination of Antioxidant Capacity TEAC Using EPR Method and ABTS (TEAC_ABTS_-_EPR_)

The experiments were performed following a procedure analogous to that described above, with DPPH replaced by an aqueous solution of the ABTS•^+^ cation radical. The ABTS•^+^ radical was generated by oxidation of ABTS with potassium persulfate (K_2_S_2_O_8_), according to the method described previously [37]. Trolox equivalent antioxidant capacity (TEAC_ABTS-EPR_) was determined using a regression equation describing the linear relationship between the percent inhibition of the EPR signal intensity (%I) and the molar amount of Trolox: %I = 1369.2C − 7.2, where %I is the inhibition, and C is the concentration of the sample. This equation was applied to calculate the antioxidant capacity of the tested samples, expressed as µmol TE per 100 mL of sample.

For all samples, the regression equation describing the linear relationship between percent inhibition (%I) of the EPR signal intensity and the sample concentration (C) was determined. Using this equation, the %I corresponding to C_100_ (the concentration equivalent to 100 mL of the sample) was calculated (I_100_). Based on the standard curve obtained for Trolox, the antioxidant capacity was determined and expressed in µmol TE per 100 mL of sample. The antioxidant capacity measured using the ABTS method by EPR spectroscopy is expressed as TEAC_ABTS-EPR_. EPR spectra were recorded using a Bruker EMX EPR spectrometer (Bruker-Biospin, Karlsruhe, Germany) using parameters described above.

### 3.6. Determination of Antioxidant Capacity Using UV-Vis Method and DPPH (TEAC_DPPH-UV-vis_)

Antioxidant capacity TEAC_DPPH-UV-vis_ was determined using the DPPH test and spectrophotometry UV-vis method [38]. UV–vis spectrophotometric measurements were performed at 515 nm using a Lambda Bio 40 spectrometer (Perkin Elmer, USA). The regression equation for the linear relationship between the percent inhibition of absorbance and the mol number of trolox was assessed as y = 1491x + 0.5, where y is the inhibition [%] and x is the volume of the sample [mL]. This equation was used to calculate the antioxidant activity in µmol trolox per 100 mL of the studied samples (µmol TE/100 mL).

The percent inhibition of the decrease in absorption at 515 nm calculated according to the following equation: % Inhibition = [(A_0_ − A)/A_0_] × 100%, where A_0_ is the absorbance of DPPH• (control sample), and A is the absorbance of DPPH• with a sample.

For all samples regression equation of linear relationship of the percent inhibition of the absorbance to the concentration of sample was determined. Using this equation, the %I corresponding to C_100_ (the concentration equivalent to 100 mL of the sample) was calculated (I_100_). Then from the standard curve the antioxidant activity µM trolox per 100 mL of sample (µmol TE/100 mL) was defined.

### 3.7. Statistical Analysis

All experiments were performed in at least three independent replicates. Data are presented as mean ± standard deviation (SD). Correlation coefficients, TPC; TEAC_DPPH-EPR_; TEAC_DPPH-UV–vis_; and TEAC_ABTS-EPR_, were calculated by the Pearson test. In addition, a one-way ANOVA test was performed to compare the mean values obtained for the different drying methods and for samples from various plant parts, both whole and crushed. The results showed *p* < 0.05, indicating that the observed differences are statistically significant.

## 4. Conclusions

The hemp plant, *C. sativa*, has emerged as a promising source of natural antioxidants, with various parts of the plant exhibiting diverse levels of antioxidant activity. Our research has revealed that extracts from *C. sativa* demonstrate potent antioxidant properties, which are significantly influenced by both the specific part of the plant used and the drying method employed. Notably, crushed plant materials showed enhanced antioxidant activity and a higher total phenolic content, indicating that preparation methods play a crucial role in optimizing the extraction of bioactive compounds.

These findings underscore the importance of both plant part selection and preparation methods in optimizing the extraction of antioxidants from *C. sativa*. They also highlight the potential of cannabis leaves, often considered a byproduct, as a valuable source of natural antioxidants, challenging the traditional view of these parts as waste. This emphasizes the need for a more holistic approach to utilizing the entire plant for its beneficial compounds.

Further research is needed to unravel the precise mechanisms behind the antioxidant effects of hemp compounds, as well as to develop and refine extraction methods for more efficient yield.

As global interest in hemp rises, a deeper understanding of its health benefits, particularly its antioxidant properties, will be essential for fully capitalizing on this plant. Research into its bioactive compounds could pave the way for innovative natural remedies and contribute to developing new products that support well-being.

## Figures and Tables

**Figure 1 ijms-26-11696-f001:**
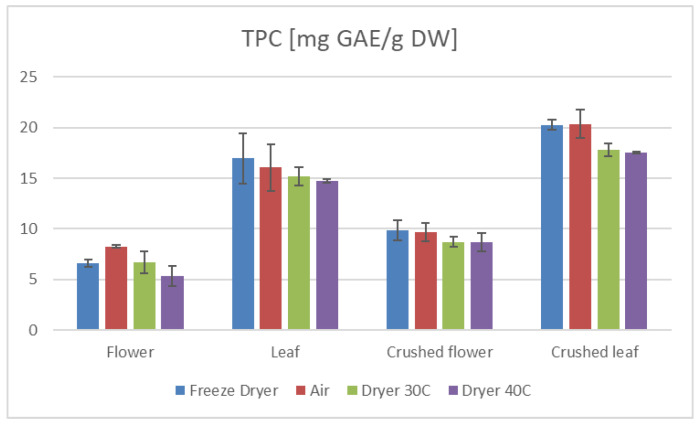
TPC results of *C. sativa* extracts presented as mg GAE/g dry mass of plant material depending on the plant part.

**Figure 2 ijms-26-11696-f002:**
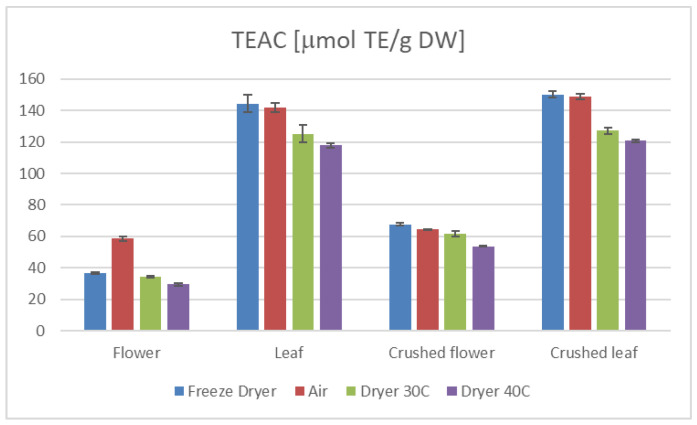
TEAC results of *C. sativa* extracts presented as μmol TE/g dry weight of plant material in the DPPH test, depending on the plant part.

**Table 1 ijms-26-11696-t001:** Total antioxidant capacity values of hemp extracts in [μmol TE/g DW].

	TEAC_DPPH-EPR_ [μmol TE/g DW]				
Plant Part	Freeze Dryer	Air	Dryer 30 °C	Dryer 40 °C	Mean ± SD
Flowers	36.6 ± 0.7	58.5 ± 1.2	34.2 ± 0.6	29.5 ± 0.9	39.7 ± 13.4
Leaves	144.4 ± 3.5	141.9 ± 3.1	125.3 ± 5.6	117.8 ± 1.5	132.3 ± 10.3
Crushed flowers	67.7 ± 0.9	64.5 ± 0.3	61.6 ± 1.6	53.6 ± 0.3	61.8 ± 3.0
Crushed leaves	150.2 ± 1.9	148.9 ± 2.0	127.2 ± 1.9	120.7 ± 0.9	136.8 ± 13.0
	**TEAC_DPPH-UV-vis_ [μmol TE/g DW]**				
Plant part	Freeze Dryer	Air	Dryer 30 °C	Dryer 40 °C	mean ± SD
Flowers	26.6 ± 0.8	49.3 ± 2.6	26.5 ± 1.1	22.5 ± 1.2	31.2 ± 3.2
Leaves	119.9 ± 1.4	118.7 ± 1.9	101.5 ± 3.6	96.1 ± 6.4	109.1 ± 9.8
Crushed flowers	57.2 ± 2.3	50.5 ± 0.7	52.9 ± 0.9	40.1 ± 2.2	50.2 ± 0.8
Crushed leaves	124.9 ± 4.2	125.7 ± 3.5	101.2 ± 4.4	98.6 ± 4.5	112.6 ± 10.2
	**TEAC_ABTS-EPR_ [μmol TE/g DW]**				
Plant part	Freeze Dryer	Air	Dryer 30 °C	Dryer 40 °C	mean ± SD
Flowers	66.3 ± 2.2	91.2 ± 0.5	43.8 ± 2.2	36.7 ± 2.4	59.5 ± 2.2
Leaves	168.8± 14.5	158.2 ± 6.7	176.8 ± 3.7	158.4 ± 6.7	165.5 ± 6.9
Crushed flowers	84.7 ± 9.6	99.1 ± 2.6	73.9 ± 2.3	65.8 ± 1.5	80.9 ± 3.2
Crushed leaves	199.4 ± 3.5	195.7 ± 4.7	160.7 ± 3.3	159.5 ± 5.9	178.8 ± 3.3

TEAC_DPPH-EPR_: Trolox equivalent antioxidant capacity determined by EPR spectroscopy using the DPPH free radical; TEAC_DPPH-UV-vis_: Trolox equivalent antioxidant capacity determined by UV-vis spectroscopy using the DPPH free radical; TEAC_ABTS-EPR_: Trolox equivalent antioxidant capacity determined by EPR spectroscopy using the ABTS free radical; Mean: Average values of antioxidant capacity of a given plant part; ±SD: standard deviation.

**Table 2 ijms-26-11696-t002:** Total phenolic content (TPC) values of hemp extracts in [mg GAE/g DW].

Plant Part	Freeze Dryer	Air	Dryer 30 °C	Dryer 40 °C	Mean ± SD
Flowers	6.6 ± 0.4	8.3 ± 0.1	6.7 ± 1.1	5.4 ± 1.0	6.7 ± 0.9
Leaves	17.0 ± 2.5	16.1 ± 2.3	15.2 ± 0.9	14.8 ± 0.2	15.8 ± 0.9
Crushed flowers	9.9 ± 1.0	9.7 ± 0.9	8.7 ± 0.5	8.6 ± 0.9	9.2 ± 0.6
Crushed leaves	20.3 ± 0.5	20.3 ± 1.4	17.8 ± 0.6	17.5 ± 0.1	19.0 ± 1.4

TPC: Total phenolic content; Mean: Average values of total phenolic content a given plant part; ±SD: standard deviation.

## Data Availability

The original contributions presented in this study are included in the article. Further inquiries can be directed to the corresponding authors.

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
