# Peer review of "Spectroscopy-Based Evaluation of the Antioxidant Capacity of Hemp (Cannabis sativa)"

_ijms, 2025, doi:10.3390/ijms262311696_

Round 1
Reviewer 1 Report
Comments and Suggestions for Authors
This paper presents an evaluation of the antioxidant properties of the leaves and flowers of a new, feminised variety of Cannabis sativa containing a Cannabis ruderalis admixture. This work employs elementary methodologies for evaluating trolox equivalent antioxidant capacity (TEAC) and total polyphenol content (TPC), in addition to electron paramagnetic resonance (EPR) and UV-vis spectroscopy. The authors emphasise the importance of preparing the material for extraction and the potential of cannabis leaves, which are often considered a by-product.
The introduction is comprehensive and the research methodology is described in detail. Preparing the material for analysis will enable the procedure to be successfully replicated.
In my opinion, the paper is well-written and well-documented, with carefully selected research methods.
Although the obtained results are not groundbreaking, they may be useful to readers in light of their scientific significance.

Author Response
Response to Reviewer 1’s report.
I want to thank you for your positive review and feedback on my article. I greatly appreciate your kind words and the time you devoted to evaluating my work.
Reviewer 2 Report
Comments and Suggestions for Authors
Figures 2 through 4 lack error bars, making it difficult to assess experimental variability.
The methodology of the drying process lacks critical details necessary to ensure the reproducibility of the results. For instance: (1) in the lyophilization process, it is stated that the product was cut into small pieces, yet no information is provided regarding the cutting method or the approximate dimensions; (2) in the natural drying process, ambient humidity conditions are not reported.
Lines 260–263. The extraction time is not mentioned. It is also not stated whether the samples underwent a size reduction or standardization process prior to extraction. From a mass transfer standpoint, it is well known that the extraction rate depends on particle size. In this case, in the absence of a standardization process, it cannot be concluded that the differences in results are due to the different extraction methods.
Tables 2 and 3 lack statistical analysis, such as mean comparisons, to determine whether there are significant differences between plant parts, between moisture reduction treatments, and between methods used to determine total antioxidant capacity. Additionally, the table numbers do not match those referenced in the main text. The first column of the table is missing a header, such as 'Plant part'.
Author Response
First of all I would like to thank you for a detailed study of my manuscript.
All Reviewers’ suggestions of the article have been taken into consideration.
Comments 1: Figures 2 through 4 lack error bars, making it difficult to assess experimental variability.
Response 1: Error bars were added to the figures as suggested by the reviewer.
Comments 2: The methodology of the drying process lacks critical details necessary to ensure the reproducibility of the results. For instance: (1) in the lyophilization process, it is stated that the product was cut into small pieces, yet no information is provided regarding the cutting method or the approximate dimensions; (2) in the natural drying process, ambient humidity conditions are not reported.
Response 2: Flowers and leaves were separated from the stem and subsequently dried.
The text in lines 242-263 has been changed according to the reviewer's comments to clarify the procedure for preparing samples for drying and analysis.
Comments 3: Lines 260–263. The extraction time is not mentioned. It is also not stated whether the samples underwent a size reduction or standardization process prior to extraction. From a mass transfer standpoint, it is well known that the extraction rate depends on particle size. In this case, in the absence of a standardization process, it cannot be concluded that the differences in results are due to the different extraction methods.
Response 3: The extraction time, water temperature, sample mass, and the size of the fragmented flowers and leaves were standardized. With the applied standardization of time, water temperature, sample mass, and the size of fragmented flowers and leaves, we believe it is possible to draw meaningful conclusions regarding the influence of different plant parts on their antioxidant properties
The following text was changed: Aqueous extracts were prepared from the dried hemp materials. Extracts were obtained from whole flowers (sample 1) and leaves (sample 2), as well as from crushed flowers (sample 3) and crushed leaves (sample 4) by pouring 50 mL of water at 90–95 °C over a 2 g sample
(lines 264-267) Aqueous extracts were prepared by pouring 50 mL of water at 90–95 °C over 2 g of dried hemp material for 15 minutes. Extracts were obtained both from whole flowers and leaves, as well as from crushed samples (particle size ~1 mm) that were sieved before extraction (see Table 1).
Comments 4: Tables 2 and 3 lack statistical analysis, such as mean comparisons, to determine whether there are significant differences between plant parts, between moisture reduction treatments, and between methods used to determine total antioxidant capacity. Additionally, the table numbers do not match those referenced in the main text. The first column of the table is missing a header, such as 'Plant part'.
Response 4: A one-way ANOVA test was performed. The results showed p < 0.05, indicating that the observed differences between the mean values obtained for the different drying methods and for samples from various plant parts, both whole and crushed are statistically significant.
We apologize for the error. The table number was entered incorrectly; it should read Table 1 instead of Table 2. A header of the first column was added.
Reviewer 3 Report
Comments and Suggestions for Authors
Comments on Manuscript ID ijms-3882854
The authors in the manuscript "Spectroscopy-Based Evaluation of the Antioxidant Capacity of Hemp (Cannabis sativa)" reported the evaluation of the antioxidant properties of various parts of the hemp plant (leaves and flowers) in a new feminised variety of Cannabis sativa with an admixture of Cannabis ruderalis, intending to give the correlation with the method of drying of the plant material.
Although the title refers to spectroscopy-based evaluation of the antioxidant potential, the EPR methods used were not commented on scientifically. Unfortunately, spectroscopy methods represented just a small part of the whole investigation, and the paper lacks a meaningful discussion regarding the advantages/disadvantages of the used EPR method in comparison to classical, and more often used, spectrophotometric methods.
Line 58, when first mentioned, the Latin plant name should be given in full – species name, author name, and family name. Afterwards, the short form should be used, for example, C. sativa. Please check throughout the text and correct.
Line 71, please uniform the terminology – vitamin E- tocopherol…and use one or another way, not both.
Lines 91, 92, the correlation between antioxidant property and polyphenolic content might be drawn, but the polyphenolic content could not be used for assessing antioxidant activity.
Table 2 – The legend is meaningless. In addition, what did the authors assume to achieve by giving the mean values of determined antioxidant capacity for different drying conditions? The aim was to point out the best drying methods and to stress the differences. Why was the ABTS test performed only using the EPR method (taking into account that the DPPH assay was performed using the EPR and UV-vis methods).
The whole section, lines 118-144, is based on comparison to the results obtained in experiments performed by other researchers, but not giving information on the methodology used, specifically as the authors stressed in the title that they employed EPR methodology to assess the antioxidant potential of the investigated samples. The rationale of the performed comparative experiments of antioxidant potential determination is lacking.
Figures 1-4 represent the repetition of the results presented in Table 2
Line 188, what does “significant” mean? The figure did not give the information on SD, there are no p-values, so it was not possible to conclude if the differences in the measurements represent significant ones.
Lines 192-194, please avoid this kind of statement, as broad, common, and not specific to the investigation performed, and not contributing to a better understanding of the obtained results.
Lines 221-223, please specify what “a slight admixture of C. ruderalis” means.
Lines 233-238, the authors gave the information that the plant material was collected on the 30th and 45th day of flowering, but the dynamic of biochemical changes was not presented in the paper.
Lines 260-262 in the results and discussion section, there is no data regarding the H2O extracts used in the investigation. If aqueous extracts were used, the author did not give the precise methodology of their preparation (Table 1 is missing).
Lines 266-266, please provide the explanation why the averages of three measurements for two different harvests were presented, when the aim was to find the best condition for obtaining the samples with the best antioxidant properties.
Line 270, please explain why, in this method, the alcoholic extract was used (compare to the statement, lines 260-263).
Line 281, 304…please give the concentration of the sample and how it was calculated.
Line 284, please, give the concentration of DPPH
Lines 289-292, 305-306… please explain the statement “the antioxidant capacity was determined in μmol TE per 100 290 mL of the sample” while the results in the tables and Figures were given in “μmol TE/g DW”.
Author Response
First of all I would like to thank you for a detailed study of my manuscript.
All Reviewers’ suggestions of the article have been taken into consideration.
General comments
Reviewer 3
General comments
The authors in the manuscript "Spectroscopy-Based Evaluation of the Antioxidant Capacity of Hemp (Cannabis sativa)" reported the evaluation of the antioxidant properties of various parts of the hemp plant (leaves and flowers) in a new feminised variety of Cannabis sativa with an admixture of Cannabis ruderalis, intending to give the correlation with the method of drying of the plant material.
Comments 1: Although the title refers to spectroscopy-based evaluation of the antioxidant potential, the EPR methods used were not commented on scientifically. Unfortunately, spectroscopy methods represented just a small part of the whole investigation, and the paper lacks a meaningful discussion regarding the advantages/disadvantages of the used EPR method in comparison to classical, and more often used, spectrophotometric methods.
Response 1: Unlike spectrophotometric assays, which rely on indirect measurements based on colorimetric or redox reactions, EPR provides direct and highly specific evidence of radical scavenging activity. This makes it a more accurate and reliable approach for evaluating antioxidant capacity, minimizing the risk of interference from non-radical reactions. The used methodology is already discussed in the published papers, and there is no need to provide an extended discussion of this issue, the citations are given.
The text in lines 96-103 has been updated with the above information.
Comments 2: Line 58, when first mentioned, the Latin plant name should be given in full – species name, author name, and family name. Afterwards, the short form should be used, for example, C. sativa. Please check throughout the text and correct.
Response 2: Corrected according to the reviewer’s comment.
Comments 3: Line 71, please uniform the terminology – vitamin E- tocopherol…and use one or another way, not both.
Response 3: Corrected according to the reviewer’s comment: the terminology has been uniformed.
Comments 4: Lines 91, 92, the correlation between antioxidant property and polyphenolic content might be drawn, but the polyphenolic content could not be used for assessing antioxidant activity.
Response 4: Corrected according to the reviewer’s comment.
Comments 5: Table 2 – The legend is meaningless. In addition, what did the authors assume to achieve by giving the mean values of determined antioxidant capacity for different drying conditions? The aim was to point out the best drying methods and to stress the differences.
Response 5: The objective of this study was to investigate the influence of drying conditions on antioxidant properties and to determine which plant part contained the highest concentration of antioxidant compounds in its aqueous extract. Furthermore, the study aimed to evaluate the impact of sample preparation of the dried plant on the efficiency of the extraction process. Accordingly, the tables present data on antioxidant capacity and polyphenol content as a function of both plant part and drying method. This approach is considered appropriate and comprehensive.
Comments 6: Why was the ABTS test performed only using the EPR method (taking into account that the DPPH assay was performed using the EPR and UV-vis methods).
Response 6: The study considered the influence of the free radical employed on the measured antioxidant properties, as well as the effect of the spectroscopic technique applied. For this reason, both the ABTS and DPPH assays were conducted using EPR, while the DPPH assay was also performed with the UV-Vis method. The ABTS assay, in particular, was carried out exclusively by EPR due to its higher sensitivity and reproducibility in detecting radical scavenging activity in the specific matrices studied. Importantly, consistent results were obtained regardless of the radical applied or the analytical technique used.
Comments 7: The whole section, lines 118-144, is based on comparison to the results obtained in experiments performed by other researchers, but not giving information on the methodology used, specifically as the authors stressed in the title that they employed EPR methodology to assess the antioxidant potential of the investigated samples. The rationale of the performed comparative experiments of antioxidant potential determination is lacking.
Response 7: This study evaluated the antioxidant capacity (TEAC) and total phenolic content (TPC) of C. sativa extracts obtained from different plant parts—flowers, leaves, and their ground forms—dried using four distinct methods. Analyses were performed using UV-Vis spectrophotometry and electron paramagnetic resonance (EPR) spectroscopy (Tables 1 and 2). The study further investigated the impact of the type of free radical on the measured antioxidant properties. Results were consistent across different radicals and methods, confirming the reliability of the measurements and enabling meaningful comparisons of antioxidant capacity across various approaches. Additionally, situating these findings within the broader literature allows for comparisons of C. sativa’s antioxidant properties with those of other plant species studied using diverse methodologies, providing a comprehensive perspective on plant-derived antioxidants. EPR spectroscopy proves to be a highly suitable and reliable method for assessing antioxidant capacity, as it directly measures the presence of free radicals, providing a more accurate reflection of radical-scavenging activity.
Comments 8: Figures 1-4 represent the repetition of the results presented in Table 2
Response 8: Indeed, the same experimental data were presented both in the figures and in the tables; however, this approach was intended to facilitate clearer visualization and to enable a more detailed assessment of the influence of different parameters on antioxidant properties. The comparative presentation was therefore chosen to highlight trends and relationships that would not be as evident if the results were shown in a single format only.
Comments 9: Line 188, what does “significant” mean? The figure did not give the information on SD, there are no p-values, so it was not possible to conclude if the differences in the measurements represent significant ones.
Response 9: Error bars have been added to the figures, and a one-way ANOVA test was performed. The results showed p < 0.05, indicating that the observed differences between the mean values obtained for the different drying methods and for samples from various plant parts, both whole and crushed, are statistically significant.
Comments 10: Lines 192-194, please avoid this kind of statement, as broad, common, and not specific to the investigation performed, and not contributing to a better understanding of the obtained results.
Response 10: The text has been removed in accordance with the reviewer’s suggestion
Comments 11: Lines 221-223, please specify what “a slight admixture of C. ruderalis” means.
Response 11: By “a small admixture of C. ruderalis” we mean the genetic contribution of Cannabis ruderalis introduced through breeding, typical for autoflowering varieties. The ruderalis genotype contribution was estimated at 5–15% based on phenotypic characteristics.
(line: 225) Information has been added to the text.
Comments 12: Lines 233-238, the authors gave the information that the plant material was collected on the 30th and 45th day of flowering, but the dynamic of biochemical changes was not presented in the paper.
Response 12: Harvest was conducted at two time points, optimal for further analysis. Material harvested between days 35 and 45 corresponds to the stage of technical maturity, characterized by stabilization of bioactive compound accumulation and preceding their potential degradation. A comprehensive assessment of the dynamics of biochemical changes is beyond the scope of this study.
Comments 13: Lines 260-262 in the results and discussion section, there is no data regarding the H2O extracts used in the investigation. If aqueous extracts were used, the author did not give the precise methodology of their preparation (Table 1 is missing).
Response 13 : The extraction time, water temperature, sample mass, and the size of the fragmented flowers and leaves were standardized. With the applied standardization of time, water temperature, sample mass, and the size of fragmented flowers and leaves, we believe it is possible to draw meaningful conclusions regarding the influence of different plant parts on their antioxidant properties
(lines 264-267) The following text was changed: Aqueous extracts were prepared from the dried hemp materials. Extracts were obtained from whole flowers (sample 1) and leaves (sample 2), as well as from crushed flowers (sample 3) and crushed leaves (sample 4) by pouring 50 mL of water at 90–95 °C over a 2 g sample
Aqueous extracts were prepared by pouring 50 mL of water at 90–95 °C over 2 g of dried hemp material for 15 minutes. Extracts were obtained both from whole flowers and leaves, as well as from crushed samples (particle size ~1 mm) that were sieved before extraction (see Table 1).
We apologize for the error. The table number was entered incorrectly; it should read Table 1 instead of Table 2. A header of the first column was added.
Comments 14: Lines 266-266, please provide the explanation why the averages of three measurements for two different harvests were presented, when the aim was to find the best condition for obtaining the samples with the best antioxidant properties.
Response 14: The averages of three measurements for the two different harvests were presented to provide representative values. The plants were at the same growth stage, and their antioxidant properties did not differ between the harvests. Therefore, this averaging approach does not affect the conclusions, as the relative differences between the conditions remain unchanged.
Comments 15: Line 270, please explain why, in this method, the alcoholic extract was used (compare to the statement, lines 260-263).
Response 15: This was an error: water extract samples of hemp were added. The mistake has been corrected, and we apologize for the oversight
Comments 16: Line 281, 304…please give the concentration of the sample and how it was calculated.
Response 16: (Line 285) C - represents the volumetric percentage of the sample added relative to the total volume (1 mL DPPH + the volume of the sample, V) - incorporated into the text according to the suggestion.
Comments 17: Line 284, please, give the concentration of DPPH
Response 17: Line 289, 200 µM DPPH solution – incorporated into the text according to the suggestion.
Comments 18: Lines 289-292, 305-306… please explain the statement “the antioxidant capacity was determined in μmol TE per 100 mL of the sample” while the results in the tables and Figures were given in “μmol TE/g DW”.
Response 18: The following text was added (lines: 296-298) and then recalculated per gram of dry matter used to prepare the extract- incorporated into the text according to the suggestion.
Round 2
Reviewer 2 Report
Comments and Suggestions for Authors
The authors have adequately addressed all comments and concerns. Therefore, I recommend accepting the manuscript for publication in its current form.
Author Response
I want to thank you for your positive review and feedback on my article. I greatly appreciate your kind words and the time you devoted to evaluating my work.
Reviewer 3 Report
Comments and Suggestions for Authors
Comments on the revised version of Manuscript ID ijms-3882854
Although improved, the manuscript still does not meet the criteria for publication in this Journal.
Table 1 - legend is not self-explanatory, as it provides the information on antioxidant properties and total phenolic content. Again, the mean values were not commented in the text, and
Lines 166-173, the results of the correlation should be presented more precisely. Please, provide information on the determined Pearson correlation coefficients between total polyphenol content (TPC) and total antioxidant capacity determined using EPR spectroscopy and UV-vis spectrophotometry with the DPPH radical (TEACDPPH-EPR and TEACDPPH-UV-vis), as well as EPR with the ABTS radical (TEACABTS-EPR) - give the specific values for each determination.
Figures 1 and 3, TPC (mg GAE/g DW), the same set of results has been presented. The same for Figures 2 and 4
Comment 12 in the previous review has not been answered, as the authors stated in their manuscript (lines 236-238): "The plants were harvested at two different times - on the 30th and 45th day of flowering - to observe the dynamics of biochemical changes occurring during inflorescence maturation".
For UV analysis in the case of DPPH assays, a concentration of 200 μM DPPH will give an absorption value higher than 1.2, and thus goes beyond the range of the Lambert-Beer law.
Author Response
Comments 1: Table 1 - legend is not self-explanatory, as it provides the information on antioxidant properties and total phenolic content. Again, the mean values were not commented in the text, and
Response 1: A detailed legend has been added to Tables (lines 120-123, 126 ) to make it self-explanatory and to clearly describe all abbreviations and parameters. A header of the first column was also added. In addition, the mean values presented in the table have now been commented on in the text (lines 161-163 ).
Line 161-163: Moreover, the average values of TEAC determined by all applied methods, as well as the total phenolic content (TPC) (Tables 1 and 2), were higher in the aqueous leaf extracts compared to those obtained for the flower extracts.
Comments 2: Lines 166-173, the results of the correlation should be presented more precisely. Please, provide information on the determined Pearson correlation coefficients between total polyphenol content (TPC) and total antioxidant capacity determined using EPR spectroscopy and UV-vis spectrophotometry with the DPPH radical (TEACDPPH-EPR and TEACDPPH-UV-vis), as well as EPR with the ABTS radical (TEACABTS-EPR) - give the specific values for each determination.
Response 2: Corrected according to the reviewer’s comment.
Line 174-179: The obtained Pearson correlation coefficients between total polyphenol content (TPC) and total antioxidant capacity determined (TEAC) are high ranging from 0.97 for TEACDPPH-Uv-vis/TPC to 0.99 for TEACDPPH-EPR/TPC and TEACABTS-EPR/TPC. It should also be emphasized that the correlation coefficients between the TEAC values ​​determined using different methods and different radicals are high and amount 0.99 for TEACDPPH-EPR/ TEACABTS-EPR, TEACDPPH-EPR/ TEACDPPH-Uv-vis and TEACDPPH-Uv-vis/ TEACABTS-EPR.
Comments 3: Figures 1 and 3, TPC (mg GAE/g DW), the same set of results has been presented. The same for Figures 2 and 4
Response 3: In accordance with the reviewer’s comment, as the data presented in Figures 3 and 4 were repetitive, these figures have been removed, and references in the text have been updated to refer to Figures 1 and 2.
Comments 4: Comment 12 in the previous review has not been answered, as the authors stated in their manuscript (lines 236-238): "The plants were harvested at two different times - on the 30th and 45th day of flowering - to observe the dynamics of biochemical changes occurring during inflorescence maturation".
Response 4: The observation of biochemical changes during inflorescence maturation is beyond the scope of this study. The sentence “to observe the dynamics of biochemical changes occurring during inflorescence maturation” was inadvertently retained in the previous version, and we apologize for this oversight. The line (236-238) has now been removed.
Comments 5: For UV analysis in the case of DPPH assays, a concentration of 200 μM DPPH will give an absorption value higher than 1.2, and thus goes beyond the range of the Lambert-Beer law.
Response 5: We appreciate the reviewer’s observation. The initial DPPH concentration of 200 μM was deliberately selected to allow simultaneous determination of TEAC by both EPR and UV–Vis spectroscopy in the same solution, ensuring comparability between the two methods. While it is true that the absorbance of DPPH at this concentration exceeds 1.2, it is important to note that upon addition of the antioxidant, the absorbance decreases to a range well within the linear domain of the Lambert–Beer law, where deviations are negligible. Moreover, the calibration curve constructed for DPPH over the relevant concentration range of trolox exhibited excellent linearity, confirming that the Beer–Lambert relationship is maintained under the assay conditions. This approach allows accurate quantification of radical-scavenging activity without compromising the consistency between EPR and UV–Vis measurements. The methodology thus ensures that TEAC values obtained from both techniques are directly comparable, while maintaining adherence to the fundamental principles of UV–Vis spectroscopy.